# Astrocytes as Neuroimmunocytes in Alzheimer’s Disease: A Biochemical Tool in the Neuron–Glia Crosstalk along the Pathogenetic Pathways

**DOI:** 10.3390/ijms241813880

**Published:** 2023-09-09

**Authors:** Stefano Stanca, Martina Rossetti, Paolo Bongioanni

**Affiliations:** 1Department of Surgical, Medical, Molecular Pathology and Critical Area, University of Pisa, Via Savi 10, 56126 Pisa, Italy; 2NeuroCare Onlus, 56100 Pisa, Italy; 3Medical Specialties Department, Azienda Ospedaliero-Universitaria Pisana, 56100 Pisa, Italy

**Keywords:** molecular anatomy of synapses, synaptic transmission, synaptic adhesion molecules, neuronal–glial networks, circuit assembly, molecular mechanisms

## Abstract

This work aimed at assessing Alzheimer’s disease (AD) pathogenesis through the investigation of the astrocytic role to transduce the load of amyloid-beta (Aβ) into neuronal death. The backbone of this review is focused on the deepening of the molecular pathways eliciting the activation of astrocytes crucial phenomena in the understanding of AD as an autoimmune pathology. The complex relations among astrocytes, Aβ and tau, together with the role played by the tripartite synapsis are discussed. A review of studies published from 1979 to 2023 on Scopus, PubMed and Google Scholar databases was conducted. The selected papers focused not only on the morphological and metabolic characteristics of astrocytes, but also on the latest notions about their multifunctional involvement in AD pathogenesis. Astrocytes participate in crucial pathways, including pruning and sprouting, by which the AD neurodegeneration evolves from an aggregopathy to neuroinflammation, loss of synapses and neuronal death. A1 astrocytes stimulate the production of pro-inflammatory molecules which have been correlated with the progression of AD cognitive impairment. Further research is needed to “hold back” the A1 polarization and, thus, to slow the worsening of the disease. AD clinical expression is the result of dysfunctional neuronal interactions, but this is only the end of a process involving a plurality of protagonists. One of these is the astrocyte, whose importance this work intends to put under the spotlight in the AD scenario, reflecting the multifaceted nature of this disease in the functional versatility of this glial population.

## 1. Background

Astrocytes, the most abundant brain non-neuronal cell type, whose processes encircle all the cellular components of the central nervous system (CNS; see a list of Abbreviations) [1], are closely connected with the surrounding neurons and blood vessels [2]. Astrocytes are marked by a pleomorphic activity in supporting the CNS neurons by building physical and neurochemical interactions with several cytotypes [3]. Due to their reactivity to the extracellular stimuli, they embody the cellular interface between the vascular compartment, brain microenvironment and synaptic junction [4]. The latter represents the functional unit of cognition on which to assess the impact of the neurodegenerative phenomena [5].

In the light of the astrocyte’s essential position in preserving the integrity of neurons and guaranteeing the appropriate and effective building of synaptic networks, it has been our choice to refer to Alzheimer’s disease (AD) as the symbol of the pathological framework of neuronal death [6,7]. AD has been historically defined as the extracellular load of amyloid-beta (Aβ) plaques and intracellular hyperphosphorylated tau protein tangles [8]. Generally speaking, Aβ has been associated with neuronal death and loss of synapses [9,10], although recently this interpretative approach to the AD pathogenesis has become more complex through the introduction of several intermediate protagonists between the disorder in the amyloid precursor protein (APP) processing and cognitive impairment [10]. With this new horizon, it has been considered of utmost importance to build a link, in terms of causality between the extra- and the intracellular environment, between the deposition of insoluble Aβ and the intracellular neurofibrillary tangles of tau protein, through studies demonstrating the anti-tau effect of an anti-Aβ therapy [11]. By the same token, in addition to Aβ plaques, particular attention has been paid to the soluble Aβ oligomers, crucial, according to the most recent studies, in promoting neuronal damage and the disruption of glutamatergic synapses [12]. Continuing this line of thinking, non-neuronal players have been investigated as paramount in AD etiopathogenesis [13]. As a consequence, the neuroinflammatory response, triggered by the proteinopathy, involving micro- and macroglia (astrocytes and oligodendrocytes), has recently taken a prominent position in the research on neurodegeneration [14].

Therefore, taking inspiration from the aforementioned background, this work aimed at assessing AD pathogenesis through the investigation of the astrocytic role to transduce the load of Aβ into neuronal death. The backbone of this review is focused on the deepening of the molecular pathways eliciting the activation of astrocytes, thus promoting their several reactive stages, crucial phenomena in the understanding of AD as an autoimmune disease.

## 2. Methods

The papers selected for this review focused not only on the morphological and metabolic features of astrocytes, but also on the latest notions about their multifunctional involvement in AD pathogenesis. Peer-reviewed papers were included if they were published within the period of 1979–2023 and written in English. Papers were excluded if they were inappropriate to the configuration of our review and/or focused on conditions not related to AD. Papers published from 1979 to 2023 on Scopus, PubMed and Google Scholar databases were included. The results were transferred to Mendeley and duplicates were eliminated. All authors screened the chosen articles and discussed the resulting data (e.g., year of publication, type of journal, language, titles, content) before writing this review. Finally, 114 papers were considered eligible. The majority of the studies were published from 2010 to 2023 (*n* = 89); of these *n* = 44 were from 2010 to 2019 and *n*= 45 were from 2020 to 2023.

In general, the chosen papers, mainly the most recent ones, identified astrocytes as active participants in AD pathogenesis and highlighted the numerous functions carried out by these cells above all in the context of the tripartite synapsis.

## 3. Results and Discussion

### 3.1. Astrocytes

Astrocytes respond to insults of different nature showing several structural and molecular changes in morphology, physiology and gene expression [1]. The most documented feature of active astrocytes is the formation of a glial scar that impedes axon regeneration [1].

Notably, astrocytes are distinguished into four main subtypes: protoplasmic, interlaminar, varicose-projection and fibrous (see Figure 1), based on the differences in their morphology and anatomical locations [15]. The only two types of astrocyte in the human brain are the protoplasmic and the fibrous types [1].

Protoplasmic astrocytes

These represent the most abundant type and are located in the cortical layers [15]. Their appearance is bushy or spongiform with approximately 5–10 primary large processes that spread radially from the soma establishing contacts not only with blood vessels [15,16] but also with the neuronal pre- and post-synaptic zones [17]. A single astrocyte in the dorsolateral striatum can interact with 50,000 synapses, while in the hippocampus interactions can surpass 100,000 [16]. This suggests that the morphological diversity of astrocytes is also related to their location within the CNS [16]. In normal conditions, these cortical astrocytes do not show glial fibrillary acidic protein (GFAP) reactivity [17]. GFAP is a cytoskeletal protein able to allow cytoskeletal reorganization in response to physiological or pathological stimuli [18].

Interlaminar astrocytes

These are almost exclusively found in higher primates and they have been localized in layer I of the cerebral cortex [15]. This kind of astrocyte has two types of processes: shorter fibers directed towards the cortex surface, and very long fibers that penetrate through the deep layers of the cortex [15].

Varicose-projection astrocytes

Similar to the interlaminar astrocytes, this typology is found only in primate brains. They are not numerous and they strongly express GFAP [15]. Unlike protoplasmic astrocytes, their varicose processes can pass through the space of neighboring astrocytes [15].

Fibrous astrocytes

In comparison with protoplasmic astrocytes, they are characterized by longer straight non-branched processes [15] displaying more GFAP immunoreaction [1]. They mostly reside in white matter [1]. Nevertheless, these processes have none of the numerous fine processes typical of protoplasmic astrocytes [1].

More recently, by using the CD44 antibody, cortical and hippocampal astrocytes have been sub-classified in astrocytes with or without long processes [17]. The first type differs from protoplasmic astrocytes because they are subpial and they express GFAP [17]. The latter show a mixed morphology and activity between the protoplasmic and the fibrous one. From a spatial point of view, it is important to describe the so-called “islands”. Every phenotype of astrocyte has its single domain, precisely its “island”, that does not overlap with other astrocytic domains [17]. Only very slight interconnections between processes are allowed: this circumstance is defined as “tiling” [19].

With pathological stimuli, astrocytes can go through significant morphologic and molecular phenotype changes, including cellular hypertrophy and upregulation of GFAP [20].

The intimate nature of astrocytes reflects, hence, their extreme sensibility to react to several stimuli [21].

### 3.2. Astrocytes, Beta-Amyloid and Tau: The Story of Alzheimer’s Disease Begins with Aggregopathy Going on to Immunopathy

In the AD murine model, astrocytes [18] have been shown to extend their processes towards Aβ deposits [18]. Astrocytes with GFAP-immunoreactivity are also associated with the so-called ghost neurofibrillary tangles (NFTs), the remnant of mature tangles after neuron death [18,22]. Furthermore, astrocytes are also able to internalize tau protein with the possibility of participating in tau propagation [18]. In AD, these cells have generally received less consideration compared to neurons and microglia. As a consequence, the functional impact of reactive astrocytes on AD pathophysiology has remained not adequately deepened [20]. Astrocyte reactivity in AD may be responsible for many pathological mechanisms such as neuroinflammation, synapse dysfunction and hypometabolism [20]. In AD, rather than observing an increase in the number of astrocytes, these cells face alterations in morphology (hypertrophy and increased number of their main processes [17] and biochemical properties: varying levels of GFAP expression, different orientation of processes toward or into Aβ deposits, degeneration or clasmatodendrosis) [20]. The latter has been observed mostly in fibrous astrocytes [17] and is the disintegration both of the distal cell processes of astrocytes, and of the proximal ones, closer to the astrocyte cell body [23]. Despite the numerous studies carried out in this regard, the mechanisms for which we find the astrocytes “embraced” with the Aβ plaques are not yet completely defined. While it is known that microglia cells relocate their cores towards the plaques, we still do not know if it is the astrocytes themselves which migrate towards the plaques or if they reorient their processes toward them, as shown in Figure 2 [17]. This detail is not devoid of importance because the architecture of individual astrocytes needs a domain, in terms of space, that must not be overstepped by the processes or bodies of other astrocytes [17]. This would lead to an alteration of their normal functionality associated with a major modification of synaptic plasticity and, eventually, to memory loss [5].

It has been mentioned that the number of astrocytes does not change in AD brains. How is it possible that the leak of the astrocyte from its state of “quiescence” is not accompanied by a proportional numerical proliferation of the cell? Several explanations have been proposed but, even though there is not a specific answer yet, according to the most accepted theories, there is actually an increase in astrocytic cells associated, at the same time, with a similar rate of cellular deaths; alternatively, after becoming AD-reactive astrocytes, these cells reach another “quiescent” state and stop proliferating [17].

It is well known that astrocytes can degrade Aβ, but this ability is strongly associated with the presence of apolipoprotein E (ApoE) [24]. This protein, in fact, enables astrocytes to find, internalize and degrade deposits of Aβ [24]. Therefore, ApoE-deficient astrocytes do not degrade Aβ and this may play a part in the AD pathogenesis. Moreover, in response to the presence of Aβ plaques, astrocytes can produce inflammatory mediators, including cytokines (e.g., tumor necrosis factor-alpha (TNF-α), transforming growth factor-beta (TGFβ), and interleukin (IL)-6), chemokines (e.g., MIP-1a, CXCL10, CCL5), complement factors (e.g., C3, C5–C9), and reactive oxygen species (ROS) [20]. These molecules are upregulated in AD and they can in part explain the astrocytes neuroinflammatory activities [20].

The astrocytic reaction to neurodegeneration can be assimilated to a medal: one face represents the pro-astrogliotic response that astrocytes can create through the production of cytokines, ROS or, more generally, of pro-inflammatory molecules; the opposite face, however, coincides with the attempt of these cells to produce neurotrophic factors, such as the brain-derived neurotrophic factor (BDNF), with the aim of protecting neurons from degeneration [25,26]. Therefore, we have on the one hand the development of a toxic environment, whereas on the other the attempt at neuroprotection. It is in this context that astrocytes, regardless of their morphology, are divided into two phenotypes: A1 and A2 [25]. A1 astrocytes are characterized by the increased expression of pro-inflammatory factors, while A2 are those that seek to reduce the harmful effects of gliosis [25]. What has not yet been clarified is what, specifically, determines the development of one phenotype rather than the other. In murine AD models, it has been observed that pro-inflammatory cytokines, such as IL-1β and IL-6, induce increased expression of APP and secretase-β [25] and, as a result, they can participate in the progression of the disease by stimulating precisely an increased formation of Aβ plaques [16].

Classical complement factors are abnormally elevated in AD brains [27]: in particular, there is a conspicuous increase in C3 expression [28], especially in the hippocampus [5]. As a result, it is logical to assume that the blockage of this factor could improve neuronal loss and brain atrophy [28,29]. Astrocytes operate some of their regulatory functions through the secretion of complement proteins: C3-factor determines microglial activation and phagocytosis of cellular debris, C8G-factor, instead, inhibits microglia activation and reduces inflammation [30]. Not coincidentally, the A1 pro-inflammatory astrocytes are characterized by an overexpression of the C3 component, thus exhibiting their potential neurotoxic effect [31]. Evidence of C3 toxicity is demonstrated by the fact that neuroprotective A2 astrocytes do not produce this molecule [31]. The presence of Aβ plaques stimulate astrocytes to produce C3 [32]. The latter is able to cooperate with the microglial C3a receptor (C3aR) constituting a communication bridge between astrocytes and microglia [32]. The cells that most express C3aR are those that are in the vicinity of Aβ deposits [32]. The result of this interaction, observed both in vitro and in vivo in murine models, reduces the microglial ability to phagocytize Aβ and consequently leads to a disease worsening [29,32].

If the intimate connection between Aβ plaques and astrocytes still has to be fully understood, the relationship with the NFTs is even more difficult to interpret. NFTs characterize the final stages of the disease and they can show both tau and GFAP reactivity [17]. Together with Aβ and NFTs, another hallmark of AD is tau. This protein, normally expressed in low levels [33], is hyperphosphorylated in AD and assembles aggregates as intraneuronal tangles [34]. Even if the presence of tau is more emblematic in other tauopathies, such as Corticobasal Degeneration and Progressive Supranuclear Palsy [34], it leads to important consequences also in AD astrocytes. Tau inclusions, internalized through lysosomal degradation, induce astrocytes to assume toxic behaviors as they make these cells responsible for its spread [34]. The disturbance of perivascular astrocytes allows tau distribution and accumulation in CNS [34].

In addition to directly interacting with neuronal cells, reactive astrocyte can directly influence vascular and perivascular cells leading to modifications in blood brain barrier (BBB) permeability [20]. In the case of neuronal damage or degeneration, astrocytes are also able to activate chemoattractant signals to recruit peripheral macrophages, white blood cells and lymphocytes into the cerebral parenchyma [20].

AD is, in this paper, examined from the perspective of astrocytes, as defining protagonists among the multiple actors in the disease. Is AD possible without the astrocyte contribution to its etiopathogenesis? A better cognitive outcome has been found by the modulation of the signal transducer and activator of transcription 3 (STAT3) [35]. It can be activated by Janus kinases (JAK) and this stimulation has been associated with AD [35]. Cytokines activate JAK and then STAT3 explicates its action at different cellular levels in astrocytes, neurons and microglia [35]. Ref. [36] Astrocytes respond to the storage of Aβ in AD with astrogliosis [35]. But if astrocytes have also been considered as a sort of guardian safeguarding the neuronal integrity and the appropriate interneuronal synaptic cross-talk, how is it possible to conceive that their peculiar reactivity to external events can be deleterious for neurons? In this regard, there is evidence that the astrocytic STAT3-pathway is affected in AD [35].

With these premises, it is possible to affirm that reactive astrocytes are more than cells involved in neuroinflammation: they are crucial active cells.

### 3.3. The Astrocyte Pathway toward Neurons in Alzheimer’s Disease

#### 3.3.1. The Tripartite Synapse: The Neuron–Glia Close Contact

There is a strict association between astrocytes (endowed with neurotransmitter receptors, such as in the hippocampus, for gamma-aminobutyric acid (GABA) [37], glutamate (Glu) [38], ATP [39] and acetylcholine [40] secreting gliotransmitters (e.g., ATP and Glu [41], and neurons in guaranteeing the adequate homeostasis of the intrasynaptic cross-talk [7]. This functional unit, based on the relationship between these three protagonists, the presynaptic neuron, postsynaptic neuron and the astrocyte, has been called the tripartite synapse [42,43]. Given the importance of astrocytes in providing a physical support to pre- and postsynaptic neurons, it is clear that it is necessary to expand the immune and supportive role of astrocytes up to their active interaction with neurons [44].

The astrocyte-neuron crosstalk in physical as well as in neurochemical terms represents an essential keystone of brain homeostasis, as a consequence of the demonstrated astrocytic excitability, relying on the increased post-stimulation levels of intracellular Ca^2+^ [7]. These Ca^2+^ fluctuations happen in the so-called Ca^2+^ microdomains consisting of a non-well delimited circuitry represented by the interaction between the perisynaptic astrocytic processes (PAPs) and pre- and postsynaptic neurons, as a result of the releasing of this cation by the endoplasmic reticule [45]. These actors, portrayed in Figure 3, represent the substrate of the bidirectional neural and glial transmission [45,46].

Astrocytes are endowed with several processes enveloping vessels, structuring the BBB integrity, affecting blood flow and permeability. Astrocytes represent, hence, a sort of interactive sleeve, interfacing dynamically the microenvironment with neurons, from the modulation of blood flow to their supportive role of synapse [47]. They are crucial in defining the cytokine microenvironment and the synaptogenesis itself. At the same time, it is necessary to underline always their prominent position in the immune system: astrocytes as antigen-presenting cells express major histocompatibility complex (MHC)-II antigens [48] and costimulatory antigens B7 and CD40 able to trigger the so-called T_h2_ response [48]. These plural functional levels, in which astrocytes are involved, mirror the complexity of a disease such as AD whose etiopathogenesis relies on an intertwining of aggregopathic [49] and autoimmune processes [50].

This is the reason why, an integrated approach aiming at remarking the immunoproteopathic nature of AD is absolutely necessary [50]. Indeed, without an immune response triggered by Aβ, characterized by a cytokine cross-talk between microglia and astrocytes, it is not possible to have AD progression [51]. Even if they do not represent the object of this study, a brief mention of microglia is necessary to frame the cellular scenario in which astrocytes are placed. Microglia, together with astrocytes, carry out multiple functions: they facilitate the synaptic communication, safeguard cerebral homeostasis and release CNS immune molecules [52]. Exactly like astrocytes, microglia are present in two different states “resting” and “activated”: the first phenotype is normally present in physiological conditions, the activated one is observable in the case of neuronal injury [53]. Microglia, by virtue of their antithetic polarization M1(pro-inflammatory) and M2 (anti-inflammatory), cover an ambivalent role in the pathogenesis of AD [14]. M1-microglia produce pro-inflammatory molecules, such as interferon (IFN)-γ, secrete inducible nitric oxide synthase (iNOS), TNF-α, IL1-β, IL-6 and chemokines [54]. These pathways are neurotoxic and, thereby, crucial in the neurodegeneration process [55]. M2 microglia, conversely, with its anti-inflammatory activity, plays a leading role in preserving neuronal homeostasis through the release of IL-4 and IL-13, glia-derived neurotrophic factor (GDNF), BDNF and in the phagocytosis of cell debris, thus safeguarding the brain microenvironment [12]. Covering a crucial position in building synapses, from the pruning to sprouting and the consequent structuring of intraneuronal connections [56], how do they project an Aβ-related direct and indirect damage to brain parenchyma into a synapse loss? Due to the tripartite nature of synapsis, the functional interaction at the basis of cognition involves, at the same time not just neurons but also glia in a cross-talk via neural and glial neurotransmitters. In this work we have investigated astrocytes as a crucial pathway by which the AD neurodegeneration evolves from neuroinflammation to neuronal death [42].

#### 3.3.2. Pruning and Adult Remodeling: Astrocytes from Normality to Alzheimer’s Disease

Pruning, the selective elimination of useless synapses [57,58] is a phenomenon in which glia lead a process of synapse selection in order to improve the effectiveness of neural networks [59,60]. In this crucial mechanism, taking place in the individual transition from childhood to adulthood, a key role is played by astrocytes [5] as well as microglia [61] although by different mechanisms [62]. Indeed, astrocyte-mediated pruning happens through multiple EGF like domains 10 (MEGF10) and MER tyrosine kinase (MERTK) pathways, two receptors able to recognize specific molecular structures exposed by degenerating neurons such as phosphatidylserine [63], whereas microglia involve the complement system to reach the same goal [64].

As highlighted in adult mice, the downregulation of the phagocytic receptors MEGF10 and MERTK [65] results, at the hippocampal level, in a decrease in the elimination of excitatory synapses causing dysregulated interactions. In fact, both pruning and adult remodeling aim at the same effect: a synapse clearance to make more effective the synaptic interactions (Figure 4). If on one hand an impaired pruning has a pathogenetic potential in neurological terms, on the other, in adult subjects affected by AD, a reduced expression of MEGF10 and MERTK has been detected [5]. Astrocytes together with microglia, by which they are influenced through IL-1 and TNF-α, are the protagonists of phagocytosis. Impaired astrocytic phagocytosis has been shown in AD murine models as well as in AD human subjects [66]. In promoting the astrocyte-related phagocytosis dysregulation, the extracellular Aβ has been imputed differently from the intracellular NFTs [67]. The postulated mechanism is, once again, the downregulation of MEGF10 and MERTK [67].

#### 3.3.3. Sprouting and the Building of New Synapses: Astrocytes from Normality to Alzheimer’s Disease

Sprouting refers to the generation of new neurites and the formation of new synaptic contacts. Even though, classically, AD has been associated with the numeric reduction of neurons and synapses, an important hallmark of the AD neurodegeneration is embodied, by the presence of dysfunctional synaptic interactions not cleared or newly built [68,69]. This interaction can take place at several levels, in physical terms by the closest contact between astrocytes and neurons and neurochemically by thrombospondin (TSP) [70]. Astrocytes involved in the phagocytosis of synapses are, at the same time, implicated in the synaptogenesis [71]. In particular, this phenomenon is affected by the secretion of specific molecules, such as neuroligins, glypican, hevin/sparc and TSP [15,72]. The astrocytic isoforms TSP1 and TSP2 are linked to the synaptogenesis, promoted by binding voltage-dependent Ca^2+^ channel subunit α_2_δ_1_ [71], as highlighted in the murine model, uncovering their expression only in immature astrocytes [73]. In particular, reduced levels of TSP1 in AD brains have been found related to an anti-Aβ and neuroprotective effect of TSP1 in AD murine models [74,75].

#### 3.3.4. Astrocytes too, Not Only Neurons: What about the Clearance of Neurotransmitters in the Synaptic Cleft in Alzheimer’s Disease

Astrocytes as cells of support to the neuroelectric activity [76] are fundamental in the Glu-glutamine(Gln) cycle promoting, by the enzyme Gln synthetase found only in astroglia [77], the silencing of the Glu-related excitatory effect with its neurotoxic potential. The concept of metabolic compartmentation has been introduced [78,79] to refer to the double cellular structure involved: neurons and astrocytes. In particular, in support of the bidirectional crosstalk as the basis of the tripartite synapse Glu has been found to be secreted not only by presynaptic neurons, but also by astrocytes through the cytoplasmic increasing of Ca^2+^ as second intracellular messenger, explicating its extracellular action binding ionotropic or metabotropic Glu-receptors (iGluR and mGluR) [78]. Glu releasing from astrocytes takes place through the cysteine-Glu antiporter, particularly sensitive to oxidative stress [80].

The involvement of astrocytes in AD affects their capacity of removing Glu from the synaptic junction, thus leading to an increased level of this neurotransmitter with excitotoxic repercussions [81]. Therefore, the astrocyte capacity to control the Glu levels in the synaptic cleft is associated not only with its introjecting Glu, but also with the conversion of Glu into Gln [81]. The Glu transporters GLT1 and GLAST are reduced in AD brains [82,83]. Conversely, the expression of the adenosine receptor A_2A_R is augmented in AD and associated with an increased releasing of Glu [84]. However, as regards A_2A_R, other studies have evidenced its protective role in AD, by virtue of its antagonizing the decreased Glu uptake associated with Aβ deposition [85].

Aβ stimulates the increased Glu secretion from neurons and astrocytes with augmented extracellular Glu levels with consequent excitotoxic effect [83]. To reach this conclusion, the first important step has been represented by the highlighting, by fluorescence studies, of the excitable nature of astrocytes in addition to their renowned reactivity to inflammatory stimuli [83]. Therefore, several studies have also investigated the possible pathogenetic role of the astrocytic iGluRs in AD, from their implications at several levels in neuroinflammation, and the secretion of cytokines [86] to the Glu-mediated excitotoxicity [87] and the cellular reshaping in astrogliosis [88]. At the same time, the treatment with antagonists of the two subunits GluN2A and GluN2B has led to an augmented Aβ neurotoxic effect, thus revealing their protective potential [89].

In particular these studies [90,91,92] have brought to light in astrocytes increased levels of intracellular Ca^2+^ spontaneously [93] or in response to the glutamatergic stimulation [94]. Therefore, astrocytes exchange information with neurons, thus becoming part of the brain plasticity process. This cytoplasmic Ca^2+^ is released by its channels on the endoplasmic reticule after the Glu stimulation through the G-protein/phosphatidylinositol-triphosphate/inositol-triphosphate [94] pathway. As already highlighted [7,95], the relevance of these depolarizations is based on the astrocyte’s extreme capacity of building interactions with up to 100,000 [96] or even 2,000,000 [97,98] synapses. So, in the light of such a rapid glance into Ca^2+^ homeostasis in the crosstalk between neural and glial transmitters, astrocytes cannot generate an action potential [99], but they are excitable towards several stimuli.

#### 3.3.5. Metabolism Impairment in the AD Astrocytes

In addition to guaranteeing brain homeostasis, pruning and synaptogenesis, astrocytes are responsible for the metabolic support of the neuronal activity determined by vasoactive molecules [100]. Therefore, in neurons, glucose metabolism, Glu secretion and vascular dilatation are deeply intertwined with astrocytes [100] and this aspect emphasizes once again the prominence of this cell population in synaptic protection and neurodegeneration.

In the onset of neurodegeneration, a glucose hypometabolism has been highlighted in astrocytes [101] where glucose is internalized through the GLUT1 transporters [102,103,104]. In this picture, astrocytes provide neuronal activity with molecules synthetized in the alveus of glucose metabolism [105]: e.g., pyruvate, the final outcome of glycolysis [106], as well as lactate in anaerobic conditions [107].

Indeed, the load of Aβ appears to be correlated to an increased glycolysis with compensatory meaning [105]. In AD, together with an augmented activity of enzymes as pyruvate kinase and lactate dehydrogenases at the frontal and temporal level [108] an emphasized 6-phosphofructo-2-kinase/fructose-2,6-biphosphatase 3 (PFKFB3) role has been evidenced as correlated to the higher expression of GFAP [109]. In this regard, an impairment to the enzymes involved in glucose metabolism appears to make astrocytes a more sensitive target for the Aβ-related cytotoxicity, as confirmed by the inhibitors of PFKFB3 [109].

Furthermore, the ApoE4, the apolipoprotein involved in the cholesterol transport, and its relative receptors, LDLR and LRP1, expressed on the astrocyte membrane, have been found to be implied in the clearance of Aβ [24,110,111]. In addition, NADPH oxidase has been seen to be important in promoting the generation of ROS and hypoxia [112]. Each of these alterations is depicted in Figure 5.

Astrocytes, as a metabolic support to neurons, embody, hence, a sounding board of the molecular changes that translates inevitably into a morphological remodeling and neurochemical adaptation.

## 4. Conclusions

Astrocytes are involved in crucial pathways by which the AD neurodegeneration evolves from an aggregopathy to neuroinflammation, then loss of synapses and neuronal death, thus resulting in the AD cognitive decline. Although neuroinflammatory processes are usually regarded as mechanisms aimed at the protection of CNS, they may acquire an active role in neurodegeneration. Recent analyses have shown that more than 50% of astrocytes in brains with neurodegenerative disorders are A1 [113]. Studies in post-mortem AD human brain tissue have confirmed the substantial imbalance between the concentration of astrocytes A1 and A2 in favour of the neurotoxic phenotype [114]. Such imbalance has been interpreted as a consequence of a greater polarization towards phenotype A1 associated with a reactivation of quiescent A1-astrocytes. As described above, the astrocytic phenotype A1 is able to stimulate the production of pro-inflammatory molecules. Elevated levels of these have been correlated with the progression of cognitive impairment and excitotoxicity derived from the increased activation of the N-methyl-D-aspartate receptor (NMDAR) [115].

Given astrocytes profusion and their multiplicity activities in AD, further investigation is needed to explore a potential role for pathology-associated astrocytes in neurodegenerative diseases. Every day scientific research expands the molecular characterization of this cell population and, consequently, the next step would be to use the recently acquired knowledge to develop alternative therapeutic strategies. Acquiring the ability to “hold back” the polarization towards the neurotoxic astrocytic phenotype might reflect in a slowing of disease progression, in optimising symptom management and improving the quality of life of these patients. Objective of this work has been to emphasize the astrocytic cell in its pleomorphic functions with respect to the neuronal network, thus highlighting the AD complex neuropathological pathways through the lens of this cell. Its most peculiar immune functions, phagocytosis, secretion of cytokines and synaptogenic factors have been closely related to the neuronal sphere which has been the main target of AD since the very beginning of the research studies.

However, even if the AD clinical expression is the result of dysfunctional neuronal interactions, this is only the end of a process involving a plurality of protagonists.

One of these is the astrocyte: its relevance in the AD scenario is strictly related to the functional versatility of this glial cytotype.

Crucial to the AD development, the astrocyte embodies a cellular type whose study in the future is undoubtedly fundamental in the search for new therapeutic targets.

This critical review has potential limitations. Our evaluations may have underestimated the big picture of the role carried out by astrocytes, because of the following: (1) our baseline scenario assumed that improvement in the understanding of the astrocytic functions would persist in future research; (2) we only evaluated the pathological context of AD, not considering the role of astrocytes in other neurodegenerative diseases; and (3) the eligible criteria for the selection of the articles, as well as the heterogeneity and range of the year of publication, may have been influenced by biases.

## Figures and Tables

**Figure 1 ijms-24-13880-f001:**
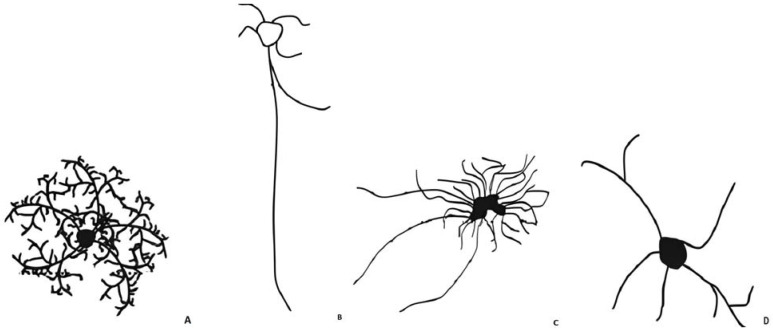
The four main astrocyte subtypes. (**A**) Protoplasmic astrocytes have bushy or spongiform appearance with approximately 5–10 primary large processes that spread radially from the soma. In normal conditions, they do not show GFAP reactivity; (**B**) Interlaminar astrocyte are almost exclusively found in higher primates and they have different types of processes: shorter fibers directed towards the cortex surface, and very long fibers that penetrate through the deep cortex; (**C**) Varicose-projection astrocytes are found only in primate brains, they strongly express GFAP and their processes can pass through the space of neighboring astrocytes; (**D**) Fibrous astrocytes are characterized by longer straight non-branched processes displaying GFAP reactivity.

**Figure 2 ijms-24-13880-f002:**
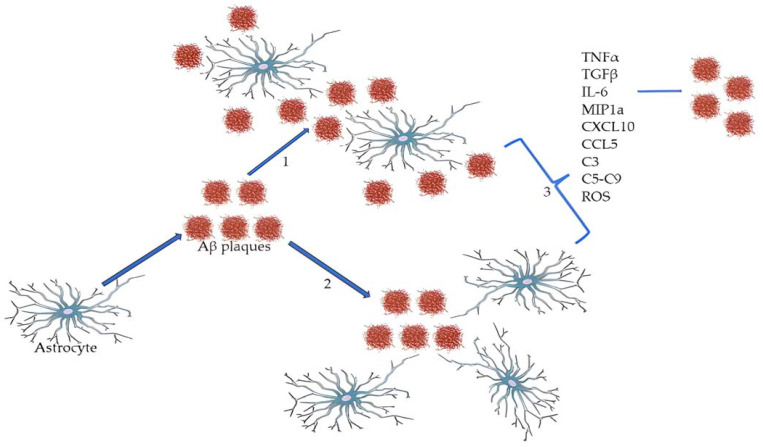
Astrocytes and Aβ interaction. The mechanisms for which we find the astrocytes “embraced” with the Aβ plaques are not yet completely defined: we still do not know if it is the astrocytes themselves which migrate and relocate their cores towards the plaques (1) or if they reorient their processes toward them (2). In both cases, in response to the presence of those aggregates, astrocytes can produce inflammatory mediators (3) that can contribute to the spread of Aβ plaques and to the development of a synaptic toxic environment.

**Figure 3 ijms-24-13880-f003:**
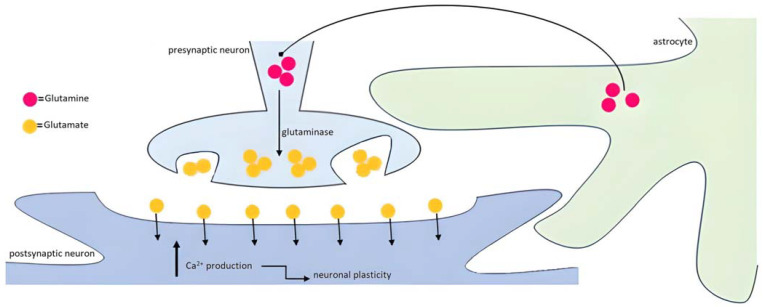
Schematic representation of the astrocyte participation in the tripartite glutamatergic synapsis. The tripartite synapse is organized by the presynaptic and postsynaptic neurons with astrocytic processes enveloping the synapses. The release of Glu from the presynaptic terminal affects the postsynaptic neuron mediating the production of intracellular Ca^2+^. Increased Ca^2+^ levels modulate, then, the neuronal plasticity.

**Figure 4 ijms-24-13880-f004:**
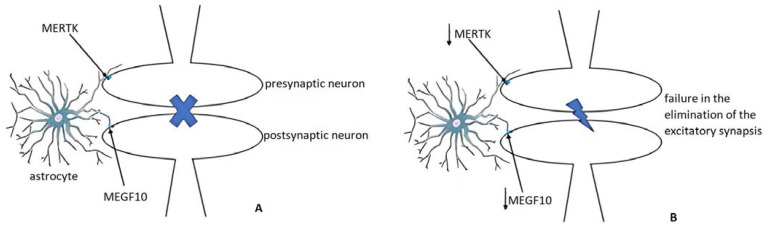
Pruning. (**A**) Astrocyte-mediated pruning physiologically happens through EGF-like domains 10 (MEGF10) and Mer tyrosine kinase (MERTK) pathways, two receptors able to recognize structures exposed by degenerating neurons; (**B**) the downregulation of MEGF10 (**↓**) and MERTK (**↓**), as detected in AD brains, results in a decrease in the elimination of excitatory synapses causing dysregulated interactions.

**Figure 5 ijms-24-13880-f005:**
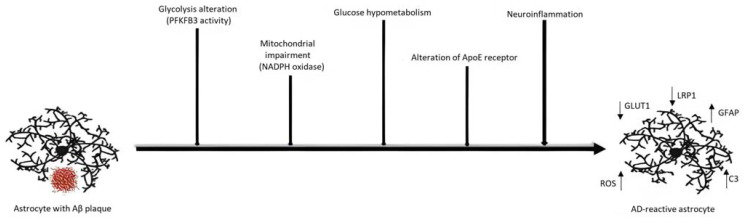
Metabolism impairment in the AD astrocytes. Numerous mechanisms have been found rearranged in AD astrocytes. The presence of Aβ plaque causes alterations in glycolysis that, through the activity of PFKFB3, provokes high expression of GFAP (**↑**). Mitochondrial impairment promotes cellular hypoxia and (**↑**) ROS production. Glucose is internalized through the GLUT1 transporters whose reduced expression (**↓**) leads to hypometabolic states. LRP1 (ApoE4 receptor) downregulation (**↓**) inhibits the expression of Aβ degrading enzymes. The neuroinflammatory environment stimulates astrocytes to produce the neurotoxic C3 factor (**↑**).

## Data Availability

Not applicable.

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
