# Peer review of "Astrocytes as Neuroimmunocytes in Alzheimer’s Disease: A Biochemical Tool in the Neuron–Glia Crosstalk along the Pathogenetic Pathways"

_ijms, 2023, doi:10.3390/ijms241813880_

Round 1
Reviewer 1 Report
The review tries to describe the role of astroglia in AD. It falls short due to (1) the organization of the manuscript; (2) English grammar and usage; (3) lack of critical evaluation of the current pieces of evidence; (4) lack of limitations of studies. Although this reviewer appreciates the author's effort in generating this review, a more comprehensive approach is needed. It is also very difficult to follow based on the use of English. As such, some sentences are ambiguous, others do not make sense.
The review tries to describe the role of astroglia in AD. It falls short due to (1) the organization of the manuscript; (2) English grammar and usage; (3) lack of critical evaluation of the current pieces of evidence; (4) lack of limitations of studies. Although this reviewer appreciates the author's effort in generating this review, a more comprehensive approach is needed. It is also very difficult to follow based on the use of English. As such, some sentences are ambiguous, others do not make sense.
Reviewer 2 Report
The text is good. However, it is difficult to understand easily, because there are no figures to explain the text. Please supplement the required figures to help for readers.
Fig. 1: Four astrocyte types features.
Fig.2: Abeta-astrocyte interaction
Fig. 3: presynaptic-postsynaptic-astrocyte interaction
Fig. 4: Pruning
Fig. 5: Metabolism of Ad-astrocyte
Minor: line 222: diriment < detrimental ?
Reviewer 3 Report
The review is generally valid. It includes results previously reported in other reviews together with others considered only marginally or left out from astrocyte functions. The presentation needs a general re-consideration to be completed also for important properties that are now incomplete or ooen to some question. Two relevant considerations need to be emphasized. The Abstract is inappropriate for many respects. Its organization is questionable, it dos not mention important data of the text, it is short (only 187 words!), after its appropriate expansion it could improve considerably. The second limitation deals with the english language. I recommend an accurate revision by collaboration with a researcher of original english literature.
As already mentioned, extensive editing of English language is required. I recommend a detailed correction of the whole text made in collaboration with a scientist of original english language.
Round 2
Reviewer 1 Report
The authors have attempted to improve this review by adding headings or subheadings and revising the English grammar and style. However, the core of the criticisms stays in place. The review does not provide any new information on the topic or is done under unbiased conditions (which would have provided new information or pointed at gaps of knowledge). The inclusion and exclusion criteria are not spelled out clearly and unequivocally. The fact that the authors met and discussed which papers were in and out without clear guidelines makes it less rigorous and with a high bias potential. It is not clear whether the papers included were also reviews, meta-analysis, etc, animals and humans, case reports etc etc. There is no indication of how the data from the included papers was extracted and summarized.
As such, the information extracted or summarized in this review could be heavily biased toward the interest of the authors (indicating that astrocytes do play a role in AD).
I commend the authors for trying to improve the English grammar and style. However, further work must be done to make the manuscript readable.
Reviewer 2 Report
I think this manuscript can be published.
